# Turbulence of Capillary Waves on Shallow Water

**Natalia Vladimirova** [1], **Ivan Vointsev** [2], **Alena Skoba** [2] **and Gregory Falkovich** [2,3,*]

[1] Department of Physics, Brown University, Providence, RI 02912, USA; nvladimirova@gmail.com
[2] Landau Institute for Theoretical Physics, 142432 Moscow, Russia; vointsevivan@gmail.com (I.V.); aoskoba@mail.ru (A.S.)
[3] Weizmann Institute of Science, Rehovot 76100, Israel
[*] Correspondence: gregory.falkovich@weizmann.ac.il; Tel.: +972-8934-2830

**Abstract:** We consider the developed turbulence of capillary waves on shallow water. Analytic theory shows that an isotropic cascade spectrum is unstable with respect to small angular perturbations, in particular, to spontaneous breakdown of the reflection symmetry and generation of nonzero momentum. By computer modeling we show that indeed a random pumping, generating on average zero momentum, produces turbulence with a nonzero total momentum. A strongly anisotropic large-scale pumping produces turbulence whose degree of anisotropy decreases along a cascade. It tends to saturation in the inertial interval and then further decreases in the dissipation interval. Surprisingly, neither the direction of the total momentum nor the direction of the compensated spectrum anisotropy is locked by our square box preferred directions (side or diagonal) but fluctuate.

**Keywords:** turbulence; capillary wave; spectrum; anisotropy





## 1. Introduction

Most of the ripples one observes on the puddles are capillary waves with wavelengths exceeding fluid depth, so they are one of the most ubiquitous waves in nature. Extensive literature on the turbulence of waves is summarized in two monographs [1,2]. As far as waves on the water surface are concerned, one finds a vast body of research on gravity waves and a substantial research on the capillary waves on a deep water. Yet surprisingly little is known about the turbulence of the capillary waves on a shallow water, except isotropic weak-turbulence spectrum and its small perturbations [1]. Needless to say that neither nature nor human activity generally provides us with nearly isotropic turbulence.

From a fundamental viewpoint it is of much interest to understand how anisotropy of forcing or geometry of the container impacts an anisotropy of developed turbulence at small scales. That problem has not been solved completely for capillary waves on a deep water either: Theory predicts that weak anisotropy imposed by forcing at large scales leads to more and more anisotropic turbulence at smaller and smaller scales, as the turbulence cascade develops [1,3]. On the other hand, initially anisotropic force-free turbulence was found numerically to undergo isotropization during turbulence decay [4], and also isotropization along the cascade was found for steady turbulence generated by a strongly anisotropic pumping [5].

This work is devoted to anisotropic turbulence of capillary waves on shallow water. The main research question that we pose is as follows: how anisotropy of the environment (forcing and/or container shape) influences the anisotropy of small-scale turbulence. Understanding far-from-equilibrium states of capillary waves on thin fluid layers in different geometries can be important, among other things, for an emerging field of liquid meta-materials—wave-driven matrices of vortices, akin to optical lattices [6]. Such wave-driven flows can give one an ability to control and separate active particles and chemicals in fluid layers in biological and engineering contexts, as well in controlled self-assembly [6,7]. The

results we present below for capillary waves on shallow liquid shed some light on this phenomenon, which turns out quite generic.

We consider developed turbulence excited by a large-scale force producing zero wave momentum. We show that the energy cascade towards small scales spontaneously breaks the reflection symmetry $\mathbf{k} \to -\mathbf{k}$ and acquires non-zero momentum. The direction of this momentum fluctuates, yet even a long-time average in a fixed reference frame demonstrates substantial anisotropy. We also elucidate connection between anisotropy and interaction non-locality in *k*-space.

## 2. Methods

Here we present the model that describes small-amplitude capillary waves on shallow fluid and the analytical methods of its statistical analysis.

### 2.1. Basic Equations

Perturbations of the fluid surface can be described by the two scalar variables: the surface elevation $\eta(x, y, t)$ and the velocity potential $\Psi(x, y, t)$. In the limit of small surface perturbations and shallow layer $\eta < h < \lambda$, the Hamiltonian has a very simple form [1]:

$$\mathcal{H} = \frac{1}{2} \int \left( \sigma |\nabla \eta|^2 + \rho(h + \eta) |\nabla \Psi|^2 \right) dx dy \tag{1}$$

Here $h$ is the depth of an undisturbed fluid in cm, $\sigma$ is the surface tension in $\text{erg/cm}^2$ and $\rho$ is the density in $\text{g/cm}^3$. The two dynamical equations of motion are

$$\frac{\partial \eta}{\partial t} = \frac{1}{\rho} \frac{\delta \mathcal{H}}{\delta \Psi} = -\text{div}\,(h + \eta) \nabla \Psi = -h \Delta \Psi - \text{div}\,\eta \nabla \Psi\,, \tag{2}$$

$$\frac{\partial \Psi}{\partial t} = -\frac{1}{\rho} \frac{\delta \mathcal{H}}{\delta \eta} = \frac{\sigma}{\rho} \Delta \eta - \frac{|\nabla \Psi|^2}{2}\,. \tag{3}$$

In the dimensionless variables, $\eta/h$, $\Psi \sqrt{\rho/(\sigma h)}$, and $t\sqrt{\sigma/(\rho h^3)}$, we obtain:

$$\mathcal{H} = \frac{1}{2} \int \left( |\nabla \eta|^2 + (1 + \eta) |\nabla \Psi|^2 \right) dx dy \tag{4}$$

$$\frac{\partial \eta}{\partial t} = -\text{div}((1 + \eta) \nabla \Psi)\,, \tag{5}$$

$$\frac{\partial \Psi}{\partial t} = -\frac{|\nabla \Psi|^2}{2} + \Delta \eta\,. \tag{6}$$

Wave amplitudes are defined as $a_{\mathbf{k}} = (\eta_{\mathbf{k}} - i\Psi_{\mathbf{k}})/\sqrt{2}$ and satisfy the equation of motion:

$$\frac{\partial a_{\mathbf{k}}}{\partial t} = -\imath k^2 a_{\mathbf{k}} - \imath \int d\mathbf{q} V_{\mathbf{kqk-q}} a_{\mathbf{q}} a_{\mathbf{k-q}} - 2\imath \int d\mathbf{q} V^*_{\mathbf{qkq-k}} a_{\mathbf{q}} a^*_{\mathbf{q-k}}\,, \tag{7}$$

which corresponds to the Hamiltonian

$$\mathcal{H} = \int \omega_k |a_{\mathbf{k}}|^2 d\mathbf{k} + \frac{1}{2} \int \left( V_{\mathbf{qpk}} a^*_{\mathbf{q}} a_{\mathbf{p}} a_{\mathbf{k}} + \text{c.c.} \right) \delta(\mathbf{q} - \mathbf{p} - \mathbf{k})\, d\mathbf{q} d\mathbf{p} d\mathbf{k} \tag{8}$$

where $\omega_k = k^2$ and $V_{\mathbf{qpk}} = p^2 + k^2 + (\mathbf{k} \cdot \mathbf{p})$. In dimensional units $\omega_k = k^2 \sqrt{\sigma h/\rho}$.

### 2.2. Simplest Properties

The system allows for 1D running wave of a permanent shape. Looking for $\eta(x - vt)$, $\Psi(x - vt)$ and introducing $z = 1 + \eta$ one obtains $v\Psi' = (z\Psi')'$, which gives the mass flux constancy $z(\Psi' - v) = V$, where the constant $V$ is the solution parameter. Now we substitute that into $2z'' = -2v\Psi' + \psi'^2 = V^2 z^{-2} - v^2$. Since we expect the profile inflection

point where the surface is undisturbed, then $z'' = 0$ when $z = 1$ and $V = v$. Integrating once, we obtain

$$z'^2/v^2 = 2C - z - 1/z.\qquad(9)$$

Integrating numerically this equation, one obtains the form of the solution for any $v$ and $C \geq 1$. The wavelength is determined by the integral

$$\frac{\lambda}{2} = \frac{1}{v}\int\limits_{C-\sqrt{C^2-1}}^{C+\sqrt{C^2-1}} \frac{dz'}{\sqrt{2C - z' - 1/z'}},\qquad(10)$$

and must be commensurate with the box size, so that $L/\lambda =$ is an integer. For small amplitude waves, $z - 1 = \varepsilon \ll 1$, which requires $C - 1 \ll 1$ we have the solution $\varepsilon = \sqrt{C^2-1}\sin(kx - k^2 t)$, that is $v = k = 2\pi/\lambda$. Existence of such solutions may hint that the system is integrable in one dimension, which can be expressed as a lack of thermalization, recurrence, etc. This is left for future studies.

## 3. Results

We now consider a set of many waves with small amplitudes. The only analytic results were obtained for the kinetic equation written for the occupation numbers defined according to $n_{\mathbf{k}} = \langle |a_{\mathbf{k}}|^2 \rangle$. In the continuous limit the equation has the form [1,2]:

$$\frac{\partial n_{\mathbf{k}}}{\partial t} = St_{\mathbf{k}} = \int d\mathbf{k}_1 d\mathbf{k}_2 \left[ U_{k12}(n_1 n_{\mathbf{k}} - n_2 n_{\mathbf{k}} - n_1 n_2) - 2U_{1k2}(n_2 n_{\mathbf{k}} - n_1 n_{\mathbf{k}} - n_1 n_2) \right],\quad(11)$$
$$U_{qpk} = \pi |V_{\mathbf{qpk}}|^2 \delta(\omega_q - \omega_p - \omega_k)\delta(\mathbf{q} - \mathbf{p} - \mathbf{k}).$$

It is valid when the interaction time $t_{NL}(k) \simeq n_{\mathbf{k}}/St_{\mathbf{k}}$ exceeds the wave period, so that the dimensionless nonlinearity parameter $\epsilon_k = 1/\omega_k t_{NL}(k) = |V_{kkk}|^2 n_k k^d/\omega_k^2 \ll 1$.

The kinetic Equation (11) conserves the energy $\int \omega_k n_{\mathbf{k}} d\mathbf{k}$ and the momentum $\mathbf{P} = \int \mathbf{k} n_{\mathbf{k}} d\mathbf{k}$. It has two known stationary isotropic solutions. The first one is the thermal equipartition $\omega_k n_k = $ const, so that in 2D the nonlinearity varies as a logarithm: $\epsilon_k = \ln(1/k)$. The second solution describes energy cascade in turbulence [1]:

$$n_k = k^{-4}.\qquad(12)$$

or in dimensional variables

$$\langle |\eta_{\mathbf{k}}|^2 \rangle = 8\varepsilon^{1/2}\sqrt{\pi}(\rho h/\sigma)^{1/4}k^{-4},\qquad(13)$$

where $\varepsilon$ is the energy flux. Such spectrum is expected to be realized in the inertial interval of wavenumbers between the pumping wavenumber $\kappa$ and the dissipation wavenumber $K$. For (12), $\epsilon_k \propto k^{-2}$, so that if nonlinearity $\epsilon_\kappa$ is small at the pumping scale, it is small everywhere. It is straightforward to check that the integrals in (11) converge on (12). Numerics demonstrate that the angle-averaged spectrum is indeed close to $k^{-4}$, as seen in Figure 1 below.

### 3.1. Analytic Theory of Angular Instability

To study the linear stability of the isotropic weak-turbulence spectrum (12), one takes $n_{\mathbf{k}} = k^{-4} + \delta n_{\mathbf{k}}$ and linearizes the kinetic equation with respect to $\delta n_{\mathbf{k}}$:

$$\frac{\partial \delta n_{\mathbf{k}}}{\partial t} = \int d\mathbf{k}_1 d\mathbf{k}_2 \left\{ U_{k12}\left[ \delta n_1(k^{-4} - k_2^{-4}) + \delta n_2(k^{-4} - k_1^{-4}) - \delta n_{\mathbf{k}}(k_1^{-4} + k_2^{-4}) \right] \right.$$
$$\left. -2U_{1k2}\left[ \delta n_{\mathbf{k}}(k_1^{-4} - k_2^{-4}) + \delta n_2(k_1^{-4} - k^{-4}) - \delta n_1(k^{-4} + k_2^{-4}) \right] \right\}.\qquad(14)$$

Spectral analysis of this equation can be found in [8], here we employ the method of moments as the most compact way to establish the instability. We denote $\theta$ the polar angle of the vector $\mathbf{k}$ and define the moments of the relative perturbation $\delta n_{\mathbf{k}}/n_k = k^4 \delta n_{\mathbf{k}}$:

$$M_m = \int_\kappa^K k^3 dk \int_0^{2\pi} \delta n_{\mathbf{k}} \cos(m\theta)\, d\theta\,, \tag{15}$$

Here $\kappa$ and $K$ are respectively pumping and dissipation scales, setting the inertial interval of the turbulence cascade. We now substitute $V_{k12}$ and $\omega_k$ into (14) and integrate it to obtain the time derivatives of the moments. The most salient point is that convergence of the integrals are different for even and odd angular harmonics. For even $m$, two cancelations at small wavenumbers and one cancelation at large ones make the integrals convergent and thus independent of $\kappa \to 0$ and $K \to \infty$. Convergent integrals give $dM_m/dt$ proportional to $m$ times a negative number (expressed via beta functions). The isotropic spectrum is thus linearly stable with respect to even harmonics. On the contrary, for odd harmonics $m = 2j + 1$, the cancelation proceed in powers of $k$ rather than $k^2$, so that logarithmic divergencies remain. With logarithmic accuracy we obtain:

$$\frac{dM_m}{dt} = (-1)^j m \ln(K/\kappa)\,. \tag{16}$$

We see that the isotropic spectrum must be linearly unstable with respect to double odd harmonics $m = 1, 5, \ldots$

### 3.2. Numerical Modelling of Strongly Anisotropic Spectra

Analysis of substantially anisotropic turbulence is done by numerical solution of system (5) and (6) in double periodic domain. Partial derivatives are computed in Fourier space, using fast Fourier transform from FFTW library. Time advancement of $\Psi$ and $\eta$, with right hand side as in (5) and (6), is done with the 4th-order Runge-Kutta scheme. Next, to provide the small-scale dissipation, we set to zero the amplitudes of one third of highest modes; this common practice of dealiasing is done in Fourier space. At the same time and also in Fourier space, we force the lower 44 modes with $0 < |k| < 4$ by adding a noise, $f\xi k^{-2}\sqrt{\Delta t}$, to the Fourier images of $\Psi$ and $\eta$, to provide forcing. Here $f$ is the force amplitude and $\xi$ is the random noise white in time and uniformly distributed in the interval $[-1, 1]$. The forcing is intended to model inhomogeneous pressure imposed on the surface by wind turbulence; such force must have zero total integral over space. Our forcing is substantially anisotropic, yet statistically invariant with respect to the transformation $\mathbf{k} \to -\mathbf{k}$, so that on average it does not produce any momentum. On the other hand, we have shown above that unstable harmonics include $m = 1$ which has non-zero momentum. The time steps are chosen based on spatial resolution, $\Delta t = 0.2(\Delta x)^2$, to resolve the frequencies of the highest modes, $\omega_k = k^2$. This is an analog of CFL condition for our dispersion relation, with empirical coefficient 0.2 determined in test simulations.

Figure 1 shows that the angle-averaged spectrum is indeed close to $k^{-4}$, as predicted by (12,13). We see that weak turbulence work well even for a sufficiently strong forcing: $f = 0.005$ corresponds to $\eta_{rms} \simeq 0.2$.

The most interesting questions are then as follows:

- Is there a spontaneous breaking of symmetry $\mathbf{k} \to -\mathbf{k}$ and generation of nonzero total momentum of the wave system?
- How does the degree of anisotropy behave as one goes along the energy the cascade to higher $k$?

Figures 2 and 3 answer the first question. Figure 2 presents a long run which includes the spectrum formation. We see that nonzero momentum spontaneously appears and persists. Its direction wanders around, but here is mostly oriented along the box sides ($\theta/\pi \simeq 0.5$) or diagonal ($\theta/\pi \simeq 0.75, 0.25$).

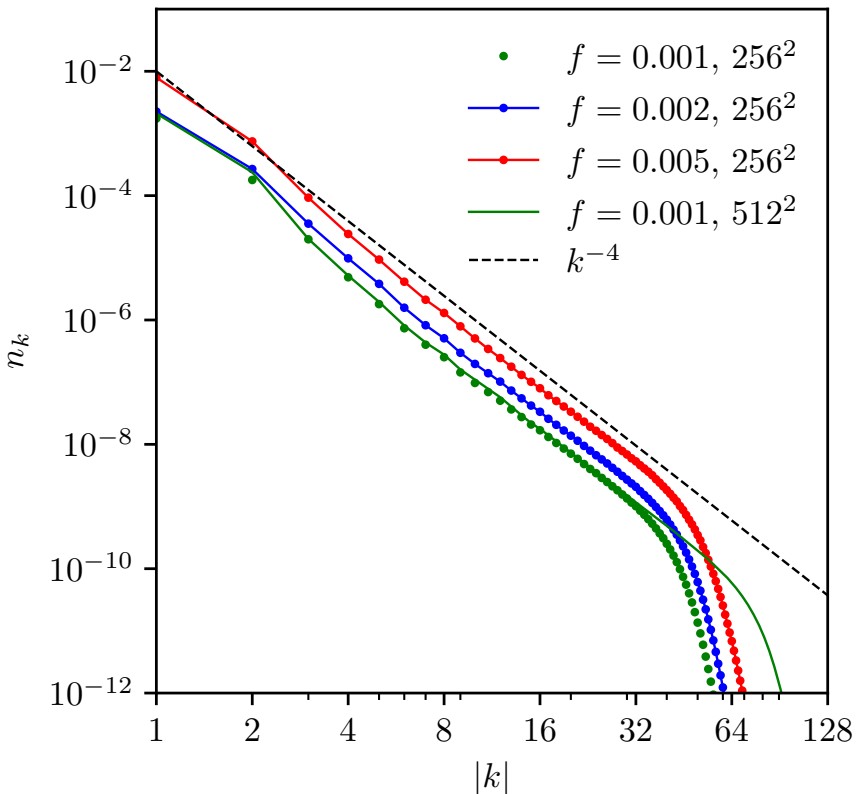

**Figure 1.** Stationary angle-averaged spectra for different force amplitudes $f$.

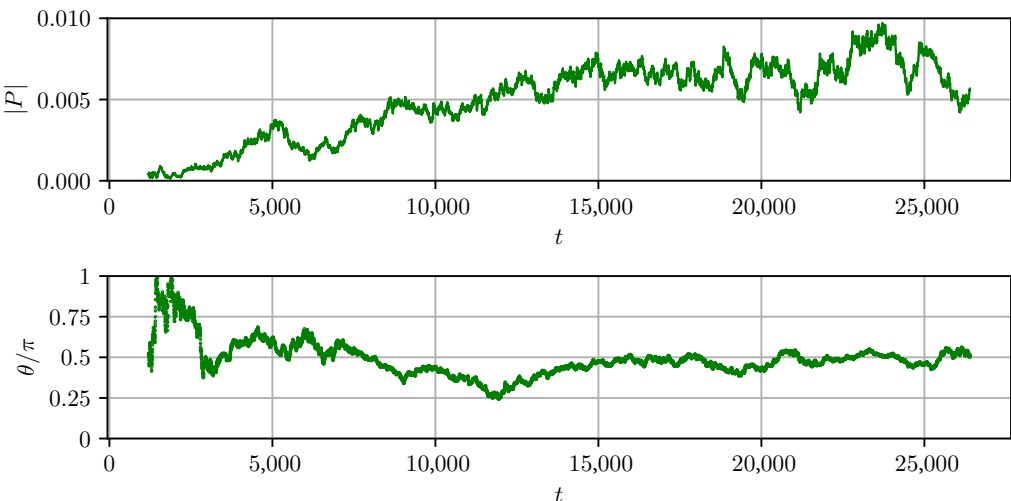

**Figure 2.** Time dependence of the momentum modulus and angle. $f = 0.001$.

Figure 3 presents the results of 10 runs starting with the same initial data of established spectrum with randomized phases. The runs are subjected to different random force realizations. For turbulence, the momentum $\sum_{\mathbf{k}} \mathbf{k} n_{\mathbf{k}}(t)$ is mostly determined by the lowest wavenumbers. To characterize the overall degree of turbulence anisotropy, we define the vector $\sum_{\mathbf{k}} \mathbf{k} k^4 n_{\mathbf{k}}$. We find that its modulus is of order unity in all cases, that is all the spectra are substantially anisotropic.

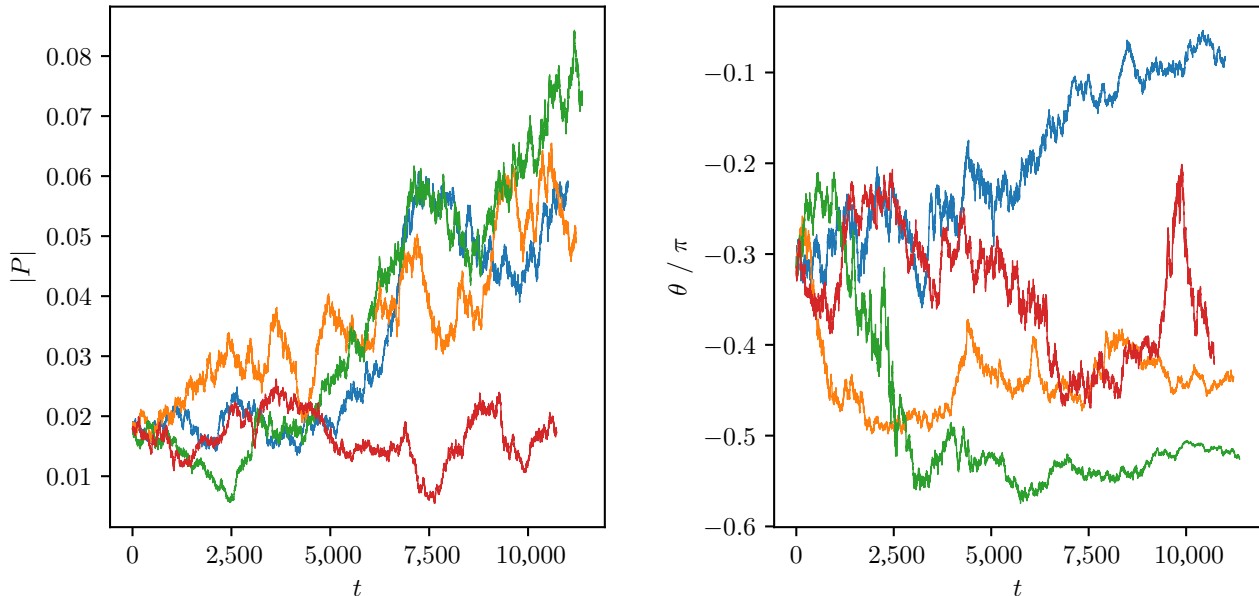

**Figure 3.** Time dependence of the momentum modulus and angle in four simulations started from the same initial spectrum, $|a_k|^2$, but having different initial phases and different realizations of the random force. $f = 0.002$.

Figures 4–6 answer the second question (on the $k$-dependence of the degree of anisotropy). Here we see that pumping-connected strongly anisotropic spectrum at low $k$ undergoes partial isotropization as one proceeds along the cascade and tend to saturate at moderate anisotropy in the inertial interval. Figure 4 shows the average over a long run (about 25,000 dimensionless times) presented in Figure 2; we have checked that averaging in the fixed reference frame gives about the same spectra as averaged in the local frame directed along the instantaneous anisotropy vector $\sum_{\mathbf{k}} k^4 n_{\mathbf{k}}(t)$.

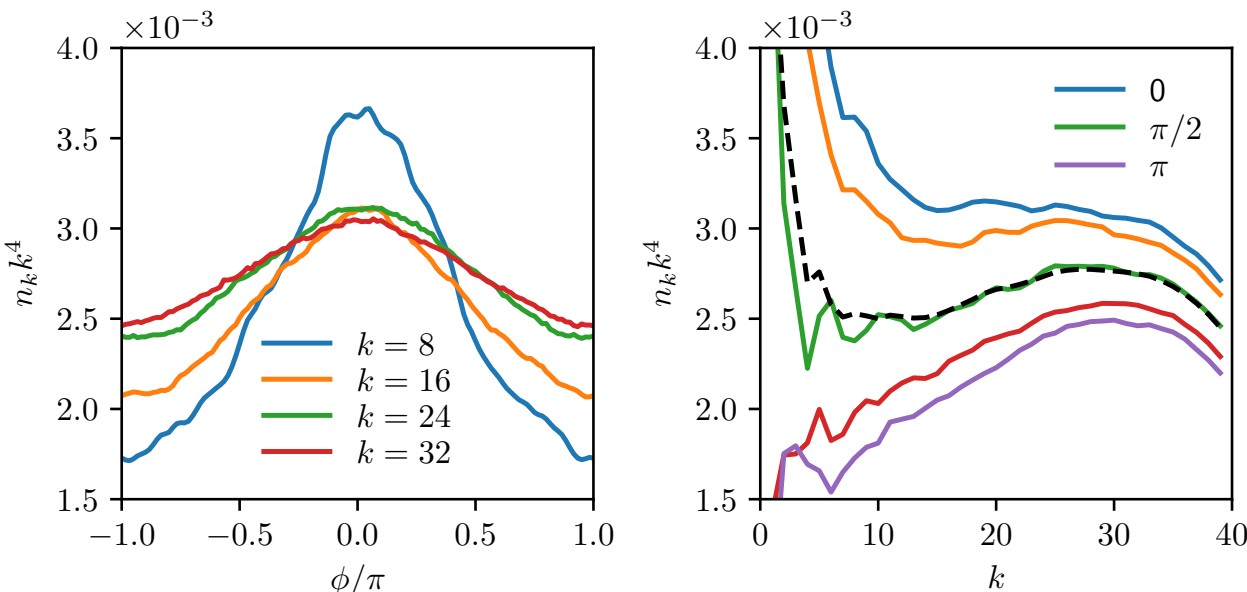

**Figure 4.** Angular (**left**) and radial (**right**) slices of the spectra averaged over a single long in the local frame along the instantaneous anisotropy vector. $f = 0.005$.

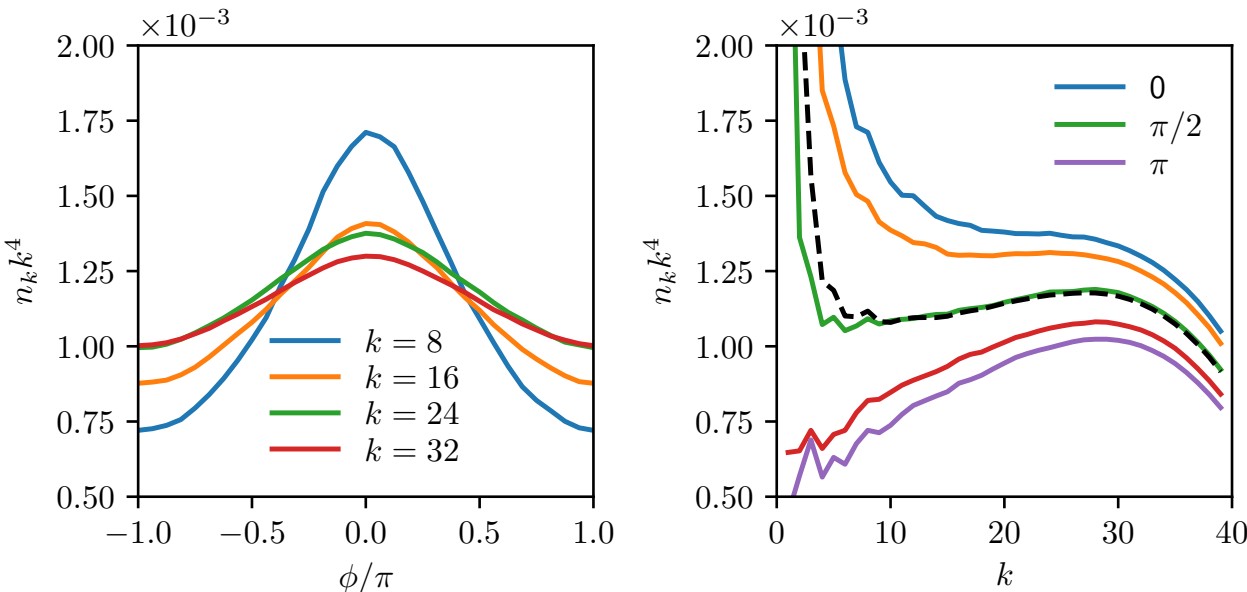

**Figure 5.** Angular (**left**) and radial (**right**) slices of the spectra averaged over 10 runs in the local frame along the instantaneous anisotropy vector, as in Figure 3. $f = 0.002$.

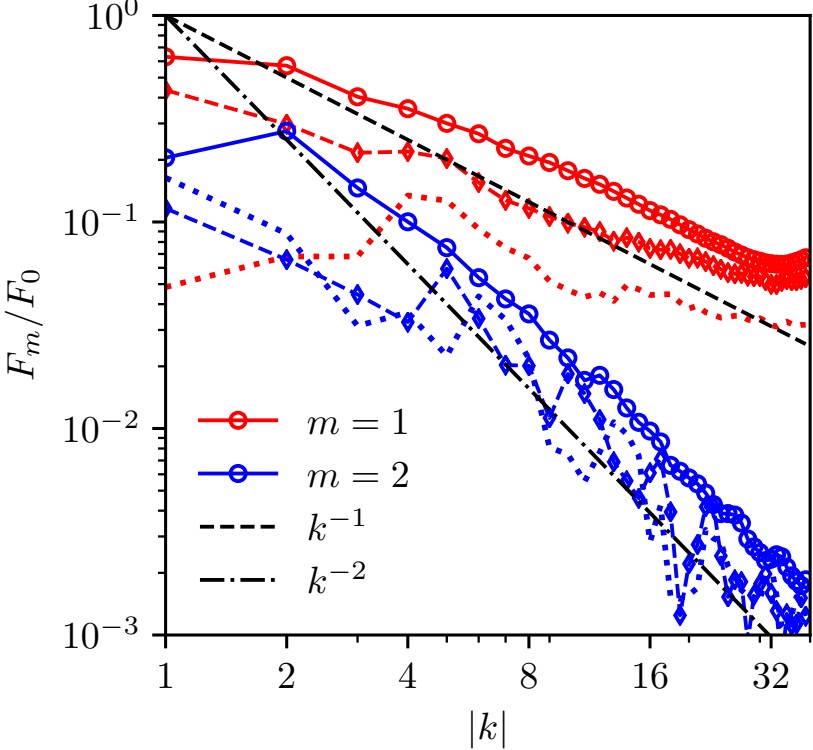

**Figure 6.** Dependence of the angular modes on $k$ for runs with $f = 0.002$. Solid lines with circles are for 10 short runs, averaged in the reference frame of the anisotropy vector, as in Figure 3. Dashed lines with diamonds are for one long run, averaged in the reference frame of the anisotropy vector, as in Figure 2; dotted lines are for the long run averaged in a static reference frame.

One obtains similar results averaging over in the local frame over 10 relatively short runs (4000 dimensionless time), all started from the same state with randomized phases, as shown in Figure 5, which is counterpart to Figure 3.

Note how the angular form of the spectrum in the inertial interval approaches the first harmonic in agreement with the theoretical prediction (16). Further, we compute the angular harmonics of the spectra

$$F_m(k) = \int_0^{2\pi} n_{\mathbf{k}} \cos(m\theta)\, d\theta \,, \tag{17}$$

Figure 6 shows the normalized first two angular harmonics for short and long averages. The data are too noisy for harmonics with $m > 2$ to be reliable. One sees that the second harmonic decays with $k$ much faster than the first one. Comparison of the dotted and dashed lines for the first harmonic shows that anisotropy in a static reference frame tend to be smaller, apparently because the direction of the anisotropy vector fluctuates in time. Yet the static average depends on $k$ only weakly, which shows that anisotropy persist in the inertial interval even on such a long timescale (over 25,000 dimensionless times).

For interpretation of Figure 6, it is instructive to recall the difference in $k$-dependence of angular perturbations to thermal equilibrium and to turbulence. Wave system with a nonzero total momentum (drift) has thermal equilibrium $n_{\mathbf{k}} \propto [\omega_k - (\mathbf{k} \cdot \mathbf{v})]^{-1}$, where $\mathbf{v}$ is the drift velocity. Therefore, the angular harmonics for weakly anisotropic spectra behave as follows:

$$\frac{F_m(k)}{F_0(k)} \propto \left( \frac{kv}{\omega_k} \right)^m \propto k^{-m} \,. \tag{18}$$

On the contrary, stationary anisotropic corrections to turbulence spectra cannot be obtained from the isotropic spectrum by the Galilean transform $\omega_k \to \omega_k - (\mathbf{k} \cdot \mathbf{v})$ [1,9]. The anisotropic correction carrying small momentum flux $\mathbf{R}$ from pumping to damping must be proportional to the dimensionless ratio of the fluxes: $F_m(k)/F_0(k) \propto (\mathbf{R} \cdot \mathbf{k})\omega_k/\varepsilon k^2 \propto k$, which grows with $k$. Since our low-$k$ pumping does not produce any momentum, we do not expect any spectral momentum flux and such behavior. Indeed, Figure 6 does not show any growth with $k$ in relative amplitude of harmonics. Surprisingly, the short-run averages behave according to (18) in the inertial interval (red and blue solid lines in Figure 6) despite this being turbulence rather than thermal equilibrium. On the other hand, long-run averages of the first harmonic do not follow these laws but rather tend to saturate (broken and dotted red lines in Figure 6.

## 4. Discussion

We first briefly discuss the applicability limits of our specific findings and then list them and present the possible wider implications. Specifically, the equations that we solve analytically and numerically describe waves with the wavelengths longer than the fluid depth yet shorter than $\sqrt{\sigma/\rho g}$, which for most fluids ranges between one and several centimeters. We described developed turbulence, that is assume a relatively wide transparency window, that is the interval of wavelength where the effects of viscous dissipation can be neglected relative to the nonlinear interaction between waves. For clean fluids (with viscosities of order $0.1 \div 0.01\,\mathrm{cm}^2/\mathrm{sec}$) even for weak nonlinearity $\eta/h < 0.1$, dissipation effects start to be important only for sum-millimeter wavelengths. However, special care is needed in applications to particle-laden surfaces, where both the surface tension and the dissipation rate may vary significantly.

Classical weak-turbulence theory gives an isotropic spectrum with local interaction and local cascade in $k$-space. However, the isotropic spectrum is unstable with respect to the anisotropic angular perturbations of double odd orders, $m = 4n + 1$, which interact non-locally both with smallest and largest wavenumbers. That instability must lead, in particular, to the spontaneous breakdown of the reflection symmetry and generation of a nonzero momentum of the wave system.

Direct numerical simulations support the prediction and show substantially anisotropic spectra without reflection symmetry. We apply an anisotropic large-scale random pumping having the square box symmetry, thus producing on average zero momentum. In all runs, the spectra spontaneously acquire nonzero momentum. Directions of the momentum and

anisotropy vectors wander with time. It may well be that averages on astronomically large timescales produce isotropic spectra; we have not been able to see any tendency towards that even on the largest timescales of tens of thousands dimensionless times.

The degree of spectrum anisotropy decreases along the cascade and tend to saturate in the inertial interval, so that the dimensionless parameter of anisotropy is of order unity, that is the anisotropy is neither weak nor strong. That may have a wider implications. For example, as mentioned in the Introduction, isotropization of small scales for decaying turbulence [4] and for a steady cascade [5] was found for capillary waves on deep water, despite theoretical prediction for linear instability with respect to angular perturbations [1,3]. It is likely that this seeming contradiction can be explained similarly to what we observe: stationary spectrum deep in the inertial has a moderate degree of anisotropy.

**Author Contributions:** Conceptualization, G.F. and N.V.; methodology, G.F. and N.V.; software, N.V.; investigation and computations, N.V., I.V. and A.S.; writing—original draft preparation, G.F. and N.V.; writing—review and editing, G.F. and N.V.; funding acquisition, G.F. All authors have read and agreed to the published version of the manuscript.

**Funding:** This research was funded by the grant 662962 of the Simons foundation, grant 075-15-2019-1893 by the Russian Ministry of Science, grant 873028 of the EU Horizon 2020 programme, and grants of ISF, BSF and Minerva. NV was in part supported by NSF grant number DMS-1814619. This work used the Extreme Science and Engineering Discovery Environment (XSEDE), which is supported by NSF grant number ACI-1548562, allocation DMS-140028.

**Data Availability Statement:** Data available upon request.

**Conflicts of Interest:** The authors declare no conflict of interest. The funders had no role in the design of the study; in the collection, analysis, or interpretation of data; in the writing of the manuscript, or in the decision to publish the results.

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
