# Peer review of "Turbulence of Capillary Waves on Shallow Water"

_fluids, doi:10.3390/fluids6050185_

Round 1

Reviewer 1 Report

The current manuscript investigates the turbulence structure of capillary waves in shallow water. It has been shown analytically and numerically that a random pumping (generating zero momentum, on average) produces turbulence with a nonzero total momentum, while an anisotropic large-scale pumping produces turbulence with an anisotropy degree that decreases along the cascade. I did not find the manuscript to be well-written and technically sound. Besides, my main problem is that the originality and contribution of the paper are limited and not satisfactory if any. 

Here are some general comments for the authors. Regarding the introduction, it does not provide sufficient background, and it is not well-written; generally, the reader is not able to draw the current state of the problem and the gaps that need to be resolved. The numerical simulation procedures are barely explained, and it lacks rigor. For example, (1) authors must be precise in describing the numerical method used; this includes assessing the formal order of accuracy of the truncation error introduced by individual terms in the governing equations. (2) Solutions over a range of significantly different grid resolutions should be presented to demonstrate grid-independent or grid-convergent results. (3) Stopping criteria for iterative calculations need to be precisely explained. Estimates must be given for the corresponding convergence error. (4) In time-dependent solutions, temporal accuracy must be demonstrated so that the spurious effects of phase error are shown to be limited. In particular, it should be demonstrated that unphysical oscillations due to numerical dispersion are significantly smaller in amplitude than captured short-wavelength (in time) features of the flow. (5) Comparison with reliable experimental results is appropriate, provided experimental uncertainty is established.

Author Response

The general doubts of the referee on the soundness, originality and contribution of our paper are not backed by any argument, so we do not address them here. It is also not clear if these doubts relate to our analytic results as well.

As far as the specific referee requests are concerned, we address them in the revised manuscript.

We substantially extended an Introduction to give a wider context for the problem we consider. The first paragraph of the Introduction is almost all new. Yet we do not think that the paper devoted to a well-defined particular subject needs to include an extended overview of the current state of the field with all its gaps. This is all the more so since there exist two detailed monographs on the subject, which we cite in the revised version.

Below, we cite the referee numbered remarks and give our responses.

(1) authors must be precise in describing the numerical method used, this includes assessing the formal order of accuracy of the truncation error
introduced by individual terms in the governing equations.

Response: Thank you for your attention to the importance of the numerical method and the errors in approximate solutions of the governing equations. Our overall method is a composition of simple discretizations described in undergraduate and graduate textbooks. Fourier transforms are used to compute spatial derivatives, and such global methods have different error characteristics than
finite difference and finite element approaches that use local (compact) support. Formally, the order of accuracy of the truncation error is the number of fourier harmonics used, in this case, 128th order accuracy, but "spectral accuracy" is used to emphasize that "order" is a misleading description. The limiting errors for spectral methods are aliasing errors: frequencies higher than those the grid can support (the Nyquist frequency) are misinterpreted by the method.
As described in the text, the highest third of frequencies supported on the mesh are set to zero amplitude; this is the standard way to ensure aliasing errors do not pollute the solution.  As for the time advancement, we do specify that we use a 4th order scheme.

(2) Solutions over a range of significantly different grid resolutions should be presented to demonstrate grid-independent or grid-convergent results.

Response: the biggest effect of the grid resolution in spectral methods is the limited extent of the spectrum, as more high-frequency modes are being excluded at coarser resolutions.  Our capillary wave
system turned out to be surprisingly forgiving the truncation -- we observe bulge at the high frequency end of the curve (bottleneck effect), and the spectra smoothly extend to the higher modes as the resolution increases. One can see it in Figure 1, in the comparison of 256^2 and 512^2 simulations.
Finer grid resolutions also require smaller timesteps, as we must resolve faster-oscillating modes. We have added a sentence about this to the revised manuscript.

(3) Stopping criteria for iterative calculations need to be precisely explained. Estimates must be given for the corresponding convergence error.

Response: There are no iterative calculations in our numerical procedure. The referee was probably confused by Eq.6 which we rewrote in the form of
$\partical \Psi / \partial t = ... $ which is more representable for numerics.

However, two legitimate questions could be asked about stopping criteria for our simulation runs in time. First, one must make sure that we have already reached a steady state. Second, runs must be long enough to gather enough statistics. To dispel possible doubts in this respect, for the revised version we made simulation runs which are three times longer than those used in the original submission. The new (three times longer) temporal evolution is presented in Fig 3. The results obtained with the expanded statistics are presented in the new Figs 5,6 - better statistics make curves smoother but does not change the results in any significant way. 

(4) In time-dependent solutions, temporal accuracy must be demonstrated so that the spurious effects of phase error are shown to be limited. In particular, it should be demonstrated that unphysical oscillations due to numerical dispersion are significantly smaller in amplitude than captured short-wavelength (in time) features of the flow.

Response: The numerical dispersion in finite difference methods originates from inability of a wave "to see ahead" beyond the size of the stencil.  This is why spectral methods, which are global, are
better suited for capturing wave dynamics.  We have tested both spatial and temporal accuracy (which are coupled because of the dispersion relation) by modeling single waves propagating in different
directions and by modeling simplified systems such as a radial wave or a combination of a small number of harmonics.  We did it with and without nonlinearity and with and without forcing.  For a single linear wave we observe correct propagation speed.  For systems with compact spectrum and no forcing we observe conservation of Hamiltonian
with relative error to 10^{-7} to 10^{-11}  per 10^5
timeseps for considered levels of nonlinearity. When, due to nonlinearity, the spectrum extends to the end of interval, the system starts losing waves and energy. This is a desired behavior, as the combination of influx at low modes and loss at high modes are essential for a statistically-steady cascade.  Finally, we have verified that forcing at different time steps result in the same level
of nonlinearity, same the averaged quantities, and the same shape of the spectrum.

(5) (5) Comparison with reliable experimental results is appropriate, provided experimental uncertainty is
established.

Response. Unfortunately, we did not find any experimental data on capillary waves on shallow water; doing controlled measurements is quite difficult in this case. This is one of the reasons that analytic theory and numerical modeling presented in our paper are of particular importance. Incidentally, the experiment inspired by this work is planned now in the Landau Institute. Excitation of waves will be done there by an anisotropic random force (computer-controlled wavemaker). Setting diagnostic takes time and will be hopefully done in a few months. We believe it is appropriate to publish the theory first.   We hope to make such a comparison in the sequel paper.

Reviewer 2 Report

The article is interesting and its structure make it quite easy to read. The main comment I have is the associated abstraction of it. It should be interesting to quote some application where the proposed development provide an improvement. There is many interesting configuration where turbulence of capillary waves are crucial and I failed to get in the article the direct connection with those application. Such a link can contribute to improve the interest of the article, to give it a little of real world. 

One other improvement could be to clarify the associated hypothesis (limit of the domain typically, caracteristics of the fluid (link with target application) ... 

Author Response

We are grateful to the referee for finding our work interesting and for making two very useful suggestions. First, in the revised version, we added to the extended Introduction an example of what we believe is one of the most promising and novel applications of surface waves: Creation of secondary flows to manipulate particles and chemicals. Second, we described in some details the applicability limits in the Discussion section, particularly addressing the above application.

Reviewer 3 Report

This is an interesting piece of work that presents relevant formal issues that need to be addressed before considering its publication:

  • Literature review is non-existent. It is said that very little is known about turbulence for capillary waves and only one reference is provided, yet a quick search reveals a significant number of works in the topic. This has to be addressed, their significance analysed and the novel aspects of this work in relation to them, highlighted. More background has to be provided.
  • English across the paper is not bad, yet it requires an in-depth review. There are some grammatical errors and, more importantly, the register at times is not at the level expected of a high-quality scientific journal.
  • The sections layout is odd.  There is no clear Results section. Also some paragraphs are misplaced. For instance, research questions are formulated within the results without been introduced previously in the Introduction. Also, new pieces of background information are given in the "Discussion", that should not incorporate new evidence besides the one provided in the results and literature review.
  • Little details are given on how the equations were solved and the discretisation was implemented. Also, I am not sure if "direct numerical simulation" is the right term to use here, since this normally applies to 3D numerical solving of Navier-Stokes without turbulence closure.

Author Response

We are grateful to the referee for the helpful suggestions. We substantially extended the Introduction, commenting more on the previous works. We cite in the revised version the two authoritative monographs on the subject, where most of the older and newer references can be found. On the other hand, we do not think a broad review of the field is needed in our relatively compact article devoted to a particular well-defined problem. The referee is right that a search on "capillary wave turbulence" brings many articles, but almost all of them are devoted to capillary waves on a deep water. Since our subject is anisotropy of turbulence of capillary waves on shallow water, we explicitly discuss the relevant works on the anisotropy of turbulence of capillary waves on a deep water. We do not wish to go beyond that and loosing the focus of the presentation. A reader that wish to have a broader picture is welcome to consult the monographs cited. In the revised manuscript, we formulate the research questions in the Introduction and also give there the background information, which was in the Discussion section before. We also added an example of possible application of capillary-wave turbulence in thin fluid layers.

We corrected several inaccuracies in English, we hope it is OK now.

We changed the layout so that the Results section is now explicitly titled. We also expanded the Discussion section adding the discussion of the applicability conditions of our results.

We have extended the Section 3.2 on numerical technique, describing the algorithm in more step-by-step approach, and rewrote equations (5)-(6) in a way more aligned with the algorithm.  We added in this Section the explanations on how the equations were solved and discretization implemented.

We agree with the reviewer that "direct" numerical simulation is a misleading term and have removed the word "direct" from description. We also made longer runs (3 times) to ensure that not only we have reached a steady state but also have enough statistics, which is reflected in new Figs 3,5,6.

Round 2

Reviewer 1 Report

The original concerns/comments of the reviewer did not get fully resolved in this revision, and I cannot recommend publishing the manuscript.

Author Response

Since the referee does not suggest any new revisions, we do not make any changes to the manuscript.

Reviewer 3 Report

I am content with the authors' revision.

Author Response

Since the referee is content, we do not make any revisions to the manuscript.